# Evaluation of miRNA Expression in Glioblastoma Stem-Like Cells: A Comparison between Normoxia and Hypoxia Microenvironment

Lucy Wanjiku Macharia [1,2], Wanjiru Muriithi [2,3], Dennis Kirii Nyaga [4], Juliana de Mattos Coelho-Aguiar [5], Tania Cristina Leite de Sampaio e Spohr [2,6] and Vivaldo Moura-Neto [1,2,*]

1. Programa de Pós-Graduação em Anatomia Patológica, Faculdade de Medicina da Universidade Federal do Rio de Janeiro (PPGAP-UFRJ), Rio de Janeiro 21941-901, Brazil; macharialw@ufrj.br
2. Instituto Estadual do Cérebro Paulo Niemeyer (IECPN), Secretaria de Estado de Saúde, Rio de Janeiro 20231-092, Brazil; cjshiru@ufrj.br (W.M.); tcspohr@gmail.com (T.C.L.S.S.)
3. Instituto de Ciências Biomédicas, Universidade Federal do Rio de Janeiro (ICB-UFRJ), Rio de Janeiro 21941-590, Brazil
4. Faculdade de Medicina, Universidade Federal do Rio de Janeiro, Rio de Janeiro 21941-901, Brazil; dennyaga@ufrj.br
5. Laboratório de Morfogênese Celular, Instituto de Ciências Biomédicas, Universidade Federal do Rio de Janeiro, Rio de Janeiro 21941-901, Brazil; jumcoelho@icb.ufrj.br
6. Centogene GbmH, 18055 Rostock, Germany
* Correspondence: vivaldomouraneto@gmail.com

**Abstract:** Purpose: Glioblastoma is an aggressive and incurable brain tumor whose progression is driven in part by glioblastoma stem cells, which are also responsible for the tumor's low therapy efficacy. The maintenance and expansion of the stem cell population is promoted by the hypoxic microenvironment, where miRNAs play fundamental roles in their survival. Methods: GBM stem-like cells were isolated from three GBM parental cell lines. The stem-like cells were then cultured under normoxic and hypoxic microenvironments followed by investigation of the in vitro "stemness" of the cells. Results: We found miR-128a-3p, 34-5p and 181a-3p to be downregulated and miR-17-5p and miR-221-3p to be upregulated in our stem-like cells compared to the GBMs. When a comparison was made between normoxia and hypoxia, a further fold downregulation was observed for miR-34-5p, miR-128a-3p and miR-181a-3p and a further upregulation was observed for miR-221-3p and 17-5p. There was an increased expression of HIF-1/2, SOX2, OCT4, VEGF, GLUT-1, BCL2 and survivin under hypoxia. Conclusion: Our data suggest that our GBMs were able to grow as stem-like cells and as spheroids. There was a differential expression of miRNAs between the stems and the GBMs and the hypoxia microenvironment influenced further dysregulation of the miRNAs and some selected genes.

**Keywords:** glioblastoma; stem cells; miRNAs; hypoxia microenvironment

## 1. Introduction

Glioblastoma (GBM) is the most aggressive cancer of the adult brain, with a median survival time of only 15 months [1,2]. Contributing to its poor prognosis are numerous therapeutic challenges, including tumor resistance and recurrence attributed mainly to the presence of a small population of tumor cells [2–5], referred to as glioblastoma stem-like cells (GSCs). GSCs reside in special hypoxic niches which are created from the inefficient blood supply that results in pseudo-palisading necrosis [6–8].

Hypoxia has been shown to drive the selection and maintenance of a stem cell phenotype [6,9,10]. Additionally, hypoxia-driven responses may enhance malignant progression and aggressiveness, resulting in increased resistance to therapy [11]. In response to low oxygen, cellular responses to hypoxia are commonly regulated by the HIFs, a family of transcriptional factors. Hypoxia-inducible factors (HIF1 and HIF2) are upregulated and

activated and consequently activate genes involved in survival mechanisms and stem cell regulation [12,13].

In an effort to identify new molecular targets for GBM therapy, studies have focused on microRNAs (miRNAs) due to their regulatory capacities in normal development and in pathological states such as cancer [14]. The biogenesis of miRNAs was explained in detail in our previous article [15]. The miRNAs play important roles in GBM and have been associated with various aspects of the GSCs selection and promotion [15,16]. However, most of the studies published report miRNAs altering a gene or a pathway that plays a role in the stemness property. For example, miR-128 has been reported to be downregulated in GBM and to target the oncogene *BMI1*, which is known to regulate proliferation, self-renewal, and migration of GSCs [17]. The miR-34 has been reported to be downregulated in glioblastoma, and it targets key stem pathways NOTCH1 and NOTCH2. Its overexpression induces apoptosis and inhibits invasion and reduces the CSC by inducing cell differentiation in glioma cells [18,19]. Ectopic expression of miR-17 has been reported to facilitate the enrichment of stem-like tumor cells in GBM cells. The cells showed an increased capacity to form colonies and neurospheres and expressed higher levels of CD133 [20]. Among the few studies reported using actual stems include a study that performed a global miRNA expression analysis, revealing the 51 most differentially expressed miRNAs between paired GSC and non-stem cell cultures. Nine miRNAs (miR-9-3p, miR-93-3p, miR-93-5p, miR-106b-5p, miR-124-3p, miR-153-3p, miR-301a-3p, miR-345-5p, and miR-652-3p) were strongly upregulated in GSCs when a comparison was made between the GSC cluster with more pronounced GSCs features and the non-stem cell cluster [21]. Similarly, a study that compared the expression levels of miR-29a in GBM cells, GSCs, human tumors, as well as normal astrocytes and normal brain by quantitative PCR, found miR-29a to be downregulated in human GBM specimens, the GSCs, and the GBM cell lines. Additionally, exogenous expression of the miR-29a inhibited GSC and GBM cell growth and induced apoptosis [22]. Together, these pieces of evidence suggest that a stem-like state in glioblastoma can influence the expression pattern of a miRNA in vitro. However, the mechanisms underlying the role of miRNAs in the stemness of GSCs are yet to be completely elucidated [23].

GBM is conventionally investigated in vitro by culturing cells as a monolayer (2D culture) or as neurospheres (clusters enriched in cancer stem-like cells) with an oxygen tension of about 20%. However, the oxygen tension in GBM in vivo ranges from 0.1% to 10% [24]. The 2D normoxic culture fails to take into account the physiologically relevant oxygen tension [25,26], as well as the interactions within the extracellular matrix, which play a role in influencing the tumor's biological properties like proliferation and motility, when investigating GBM growth in vitro [27]. Neurosphere culture has been regarded as the gold standard model for isolation of the stem cell populations. However, its main limitation is that of allowing cells to form their own niche, where more differentiated cells are positioned towards the center rather than on the surface, as well as containing a mixed population of cells and a small number of true stem cells [28]. To circumvent this limitation, conducting experiments under the hypoxia microenvironment or the use of 3D culture models may help to give accurate accounts of GBM tumor biology in vitro. This is because the two are able to mimic the intra-tumoral microenvironment more accurately than monolayer cultures performed under normoxia cultures. Therefore, we used hypoxia to study GSCs in depth. We also explored the 3D culture model as a cheaper culture alternative that is able to mimic the tumor's microenvironment.

## 2. Materials and Methods

### 2.1. Cells and Culture Conditions

Three cell lines, namely, GBM03, GBM95, and T98G, were used in this study. The GBM03 and GBM95 cell lines were prepared from biopsy samples obtained from Brazilian patients who had recurrent glioblastoma. The GBM03 patient had received radiotherapy treatment before the sample was obtained, while GBM95 had not received any form of therapy. The cell lines were established and characterized in our laboratory as described in

detail by Faria and colleagues [29]. The T98G cell line (CRL-1690) was obtained from ATCC and it was obtained from a 61 year old Caucasian male with glioblastoma. The GBM cells were maintained in Dulbecco's modified Eagle's medium (DMEM/F12) supplemented with 10% fetal bovine serum (FBS). The GBM stem-like cells (oncospheres) were obtained by first growing the GBM cells (GBM03, GBM95, and T98G) in DMEM/F12 with the fetal bovine serum. When the cells reached a semi-confluence, the old medium was removed and the culture plates were rinsed three times with 1% PBS before adding the serum-free NS34 stem cell medium made of 5x DMEM/F12 (Dulbecco's Modified Eagle Medium/Nutrient Mixture F-12), 30% glucose, 200 nM Glutamine, 1 M Hepes, 7.5% sodium bicarbonate, 100 µg/mL penicillin and streptomycin and the N2 (100×), B27 (50×) and G5 (100×) growth factors as described in detail by [30]. The cells were maintained in a normoxia chamber set at 37 °C in an atmosphere containing 95% air and 5% $CO_2$. The cells were maintained in the NS34 medium throughout the culture period. After 15 days in culture, the majority of the cells had formed oncospheres that were floating on the medium; these oncospheres were transferred into new plates to separate them from the adhered or differentiated cell lines. The oncospheres were maintained in NS34 medium and later evaluated for the expression of the transcription factors SOX2 and OCT4 and for their clonogenic ability, all indicators of an undifferentiated stem-like state. For the hypoxia experiments, the cells were cultured in a hypoxia chamber (Thermo scientific, Waltham, MA, USA) set at 1% $O_2$.

### 2.2. 3D Culture Model

The 3D cultures were performed using agarose gel as described in detail by Tang and colleague [31]. The 24- or 96-well plates were coated with either 300 µL or 100 µL of 1.5% agarose diluted in 1×PBS respectively. The agarose-coated plates were then placed in a freezer for about 5 min to allow for the agarose to solidify before placing them at room temperature before plating. For 3D cultures, only the GBM03 and T98G cell lines were used. We plated 8000 cells in each well of the 24- or 96-well plate with either the NS34 stem-cell medium or with the DMEM/F12 with 10% FBS. The cells were maintained in a chamber set at 37 °C in an atmosphere containing 95% air and 5% $CO_2$ and followed up for 12 days.

### 2.3. Clonogenicity Assay

For the clonogenicity assay, the oncospheres (stem-like cells) were transferred into a falcon tube and dissociated by pipetting up and down multiple times. The stem-like cells were counted and plated in a 96-well plate at a density of 1–2 cells per well in 100 µL of NS34 medium. The cells were incubated at 37 °C in a humidified 5% $CO_2$ and 95% air atmosphere. After a day, the plates were observed for sphere growth, and the numbers of wells with spheres was highlighted and monitored over a period of four weeks. Each week, the number of spheres in the highlighted wells was counted and noted. When necessary, 10 µL of the NS34 medium was added weekly per well to supplement the old medium. At the end of the 4 weeks, an average of the number of spheres per well was obtained, and this was also done for the biological replicates. A growth curve was drawn reflecting the self-renewal or the clonogenic ability of the cells.

### 2.4. Immunocytochemistry

Before immunocytochemistry experiments, the coverslips were treated with polylysine or polyornithine to help the oncospheres (stem-like cells) or the spheroids to adhere to the coverslips. The coverslips were placed into 24-well plates and covered with 400 µL of 1% polylysine or polyornithine then incubated for 40 min at 37 °C. Afterwards, the polylysine or polyornitin was removed from the coverslips and rinsed two times with distilled water. The plates with the ready cover slips were stored in the fridge in 1% PBS until ready for use. On the day of platting, the PBS was removed and about 20 µL of the medium containing the oncospheres or spheroids was pipetted into each well followed by 40 min of incubation at 37 °C. Later, the coverslips were rinsed once with 1% PBS. Fixing

was done using 4% PFA followed by rinsing using 1% PBS. Permeabilization was done using 0.1–1.5% triton depending on the antibody used. The slips were then incubated with 5% BSA diluted in PBS for 30 min to minimize non-specific attachments. The cells were incubated with primary antibodies including; anti-Sox2 (Cell signaling, Danvers, MA, USA, 1:400), anti-Oct4 (Cell signaling, 1:400), anti-HIF-1alpha (Millipore, Burlington, MA, USA, 1:500), anti-HIF-2alpha (Millipore, 1:500), anti-Glut-1 (Millipore, 1:500), anti-VEGF (Abcam, Cambridge, UK, 1:500), anti-Bcl2 (Abcam, 1:500) and anti-Survivin (Abcam, 1:500). On the following day, the cells were rinsed with 1% PBS and incubated with secondary antibodies conjugated with Alexa Fluor 488 (goat anti-mouse/rabbit; 1:2000) or Alexa Fluor 546 (goat anti-rabbit; 1:2000) for 2 h. This was followed by rinsing with 1% PBS and staining with DAPI, a nuclear marker. The coverslips were rinsed again in 1% PBS and then mounted on glass slides using Fluoromount-G anti-fading agent (Emsdiasum, Hatfield, PA, USA). Negative controls stained with rabbit or mouse IgG antibodies were included. The cells were imaged using DMi8 advanced fluorescence microscopy (Leica Microsystems, Wetzlar, Germany). The fluorescence intensity of individual cells was measured and analyzed using ImageJ software (NIH, Bethesda, MD, USA).

### 2.5. qPCR

Total RNA was extracted from the cells using Trizol Reagent (Ambion, Waltham, MA, USA, Life Technologies, Waltham, MA, USA) and the PureLink® RNA Mini Kit (Invitrogen, Waltham, MA, USA, Thermo Scientific) following the manufacturer's instructions. Sample RNA purity was estimated using a Nanodrop lite (Thermo scientific) spectrophotometer. One microgram of the total RNA, pool RT microRNA primer (custom made), and SuperScript™ III Reverse Transcriptase (SS III, Invitrogen, Life Technologies) was used to perform the cDNA synthesis. Quantitative real-time PCR (qRT-PCR) was performed using Power SYBR green PCR master mix (Applied biosystems, Thermofisher Scientific) and custom-made microRNA primers from Integrated DNA Technologies (IDT). The primers used were designed as shown in Table 1. The run was carried out using CFX96 Real-Time System (Bio-Rad, Hercules, CA, USA). Melt curves were included in every run and the assays were performed in triplicate. Endogenous RNA U6 (RNU6) was used as a control for the normalization of mature miRNAs. Analysis of the miRNAs expression was calculated using the $2^{-\Delta\Delta Cq}$ method [32]. The data were obtained from three independent experiments and analyzed using GraphPad Prism software version 5.

**Table 1.** Primers and sequences. Oligonucleotide primers for qPCR. "F" and "R" indicate forward and reverse sequences, respectively.

| Name | Forward Primer (5′-3′) | Reverse Primer (5′-3′) |
|---|---|---|
| miR-34-5p | ACACTCCAGCTGGGTGGCAGT GTCTTAGCT | CTCAACTGGTGTCGTGGAGTCG GCAATTCAGTTGAGAGACAACC |
| miR-128-3p | ACACTCCAGCTGGGTCA CAGTGAACCGGTC | CTCAACTGGTGTCGTGGAGTCG GCAATTCAGTTGAGAGAAAGAGAC |
| miR-221-3p | ACACTCCAGCTGGGAGC TACATTGTCTGCT | CTCAACTGGTGTCGTGGAGTCG GCAATTCAGTTGAGAGGAAACCCA |
| miR-17-5p | ACACTCCAGCTGGGCAAA GTGCTTACAGTG | CTCAACTGGTGTCGTGGAGTCG GCAATTCAGTTGAGAGCTACCTGC |
| miR-181a-3p | ACACTCCAGCTGGGAC CATCGACCGUUGAT | CTCAACTGGTGTCGTGGAGTCG GCAATTCAGTTGAGAGGGTACAAT |
| SNU6 | GCTTCGGCAGCACAT ATACTAAAAT | CTCAACTGGTGTCGTGGAGTC GGCAATTCAGTTGAGAGCGTTCCA |

### 2.6. Statistical Analysis

The experimental data are expressed as the mean $\pm$ standard deviation (SD) of three independent replicates. Statistical differences between two groups were evaluated by the

Students t-test. Those between more than two groups were subjected to analysis of variance (ANOVA) with Dunnett's as a post-test using GraphPad Prism 5.0 statistics software (GraphPad Software, Inc., La Jolla, CA, USA). $p < 0.05$ was considered statistically significant.

## 3. Results

### 3.1. Clonogenic Capacity of Established GSCs

The preparation and isolation of GSCs is a multistep process that involves oncosphere formation, followed by evaluation of the self-renewal property and the expression of stem-markers like SOX2 and OCT4 [33,34]. In this manner, GBM03, GBM95, and T98G were cultured in NS34 stem-cell medium, and the conversion process followed for five weeks. Our cells were able to form oncospheres as early as 14 days and were floating by the end of the third week (Figure S1). Notably, the oncospheres (O) formed from GBM03 and GBM95 (Figure S1A,B) were larger in diameter than the ones observed in T98G, which were also fewer in number and the slowest to convert (Figure S1C). We performed the clonogenic assay to examine the ability of a single cell to grow into a large colony through clonal expansion, also a necessary indicator of an undifferentiated cancer stem-like state [35,36]. Our stem-like cells GBM03-O and GBM95-O were seeded at a density of 1–2 cells per well in a 96-well plate in 100 μL of NS34 and followed up for a period of four weeks. We were able to observe the colony-forming ability of both the GBM03-O and GBM95-O cell lines (Figure 1A) and an oncosphere size increment along the four weeks of culture (Figure 1B,C). The GBM03-O had oncospheres that were larger in diameter when compared to the GBM95-O, demonstrating potentially higher clonogenic ability (Figure 1A–C). The clonogenic capacity of T98G has been reported previously and is in line with our data [37,38].

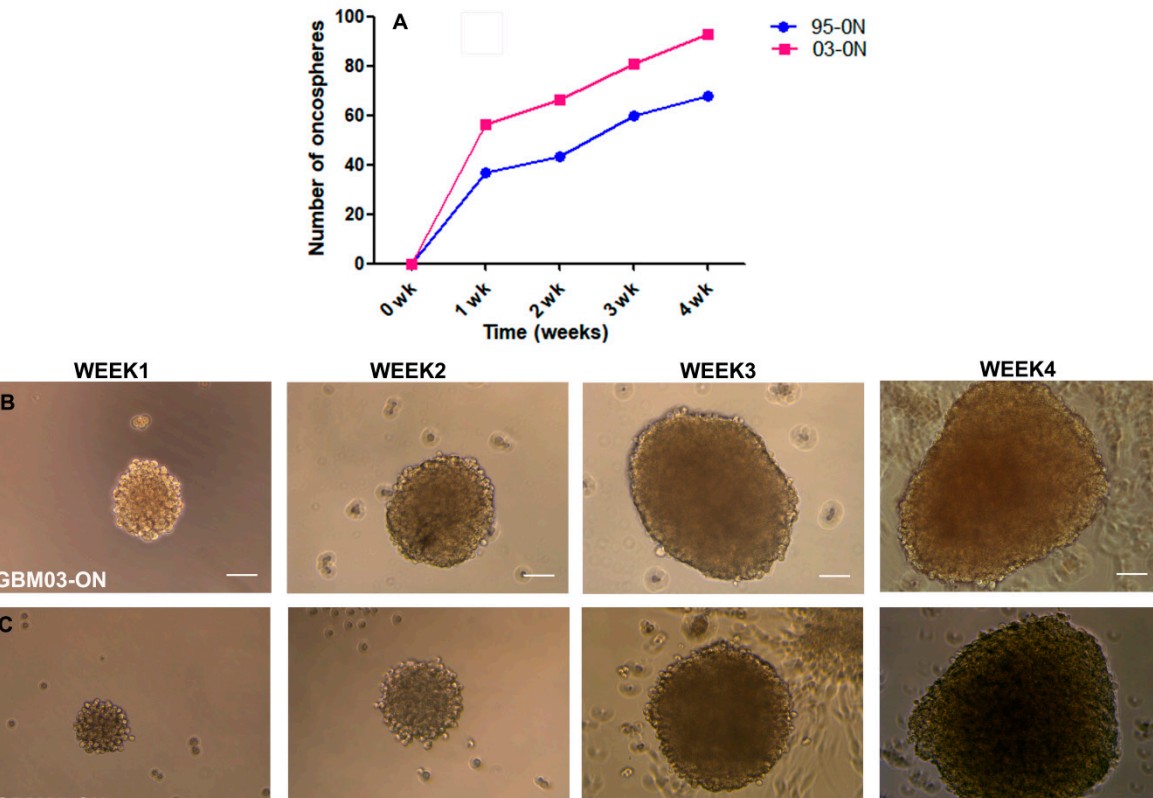

**Figure 1.** Clonogenic capacity of established GSCs. The stem-like cells isolated from GBM03 and T98G cell lines were evaluated for their self-renewal ability. (**A**) The colony formation ability of the stem-like cells over a period of four weeks. (**B,C**) The gradual size increment of the clones during the four weeks. The photos were taken using a DMi8 Leica microscope and edited using image J. Each value represents the mean ± SD of three independent runs. Scale bar = 200 μm.

### 3.2. The Expression of miRNAs by the Glioblastoma Stem-Like Cells

The miRNAs play key roles in GBM pathogenesis, but the role they play in stem cells is yet to be completely elucidated [23]. Therefore, we evaluated the miRNA expression between our stem-like cells (GBM-03-0, 95-0, and T98G-0) and their respective parental cells (GBM03, 95, and T98G) under normoxia conditions. Our results showed significant downregulation of miR-34-5p, miR-181a, and miR-128-3p (Figure 2A–C) and a significant upregulation in miR-17-5p and 221-3p (Figure 2D,E) in the stem-like cells compared to their respective parental GBMs. The expression pattern of the miRNAs was similar to when comparing the GBM cell lines with a non-GBM control cell line (results not shown).

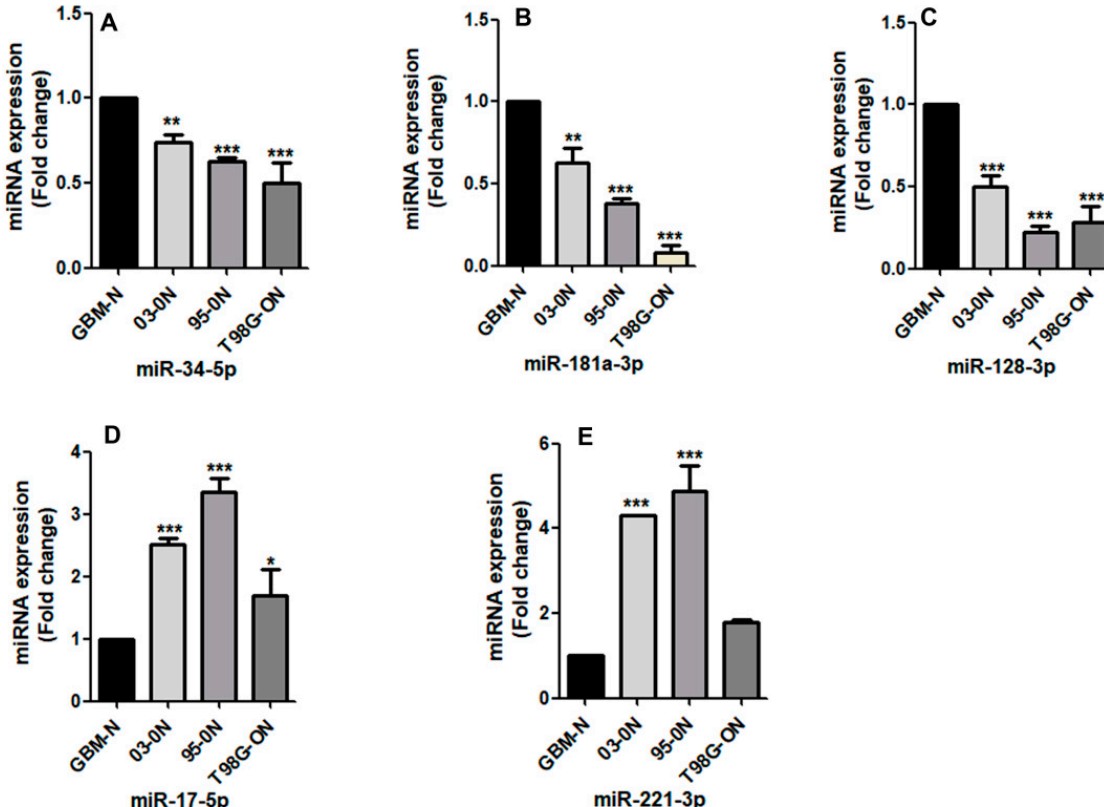

**Figure 2.** Expression of miRNAs in GBM stem-like cells by qRT-PCR. The stem-like cells (03-O, 95-O, and T98G-O) were evaluated for the expression of miR-34a (**A**), miR-181a (**B**), miR-128-3p (**C**), miR-17-5p (**D**) and miR-221-3p (**E**) by qRT-PCR and the results compared to their parental cells. The experiments were performed under normoxia microenvironment and the relative fold change difference was calculated using 2-$^{\Delta\Delta}$Cq method, where RNU6 was used as the normalizer. Each value represents the mean $\pm$ SD of three independent runs where, * $p < 0.05$, ** $p < 0.01$, *** $p < 0.001$.

### 3.3. The Effect of Hypoxia Microenvironment on the Stem-Like Cells

The hypoxia microenvironment has been associated with the reprogramming and maintenance of the stem-like cell population [6,39]. Most studies, however, investigate oncospheres in vitro by culturing cells in normoxia conditions of 21% $O_2$. Culturing our stem-like cells in a hypoxic microenvironment would help to recapitulate the oxygen tension found in a tumor microenvironment [24]. To achieve this, we cultured our stem-like cells (03-O, 95-O, and T98G-O) under a hypoxia microenvironment (1% $O_2$) for 72 h and compared their morphological changes to their counterparts grown under a normoxia microenvironment. We were able to observe that, under hypoxia, our stem-like cells (03-O, 95-O, and T98G-O) had spheres that were bigger in diameter than those of their counterparts in normoxia. Additionally, the 03-O and 95-O oncospheres had larger spheres than T98G-O (Figure S2). We also evaluated the expression of the hypoxia-inducible factor 1 and 2 (HIF-1/2$\alpha$). Our results showed an upregulated expression of HIF-1$\alpha$ in cells cultured

under hypoxia (Figure 3A–C). There was a slight expression of HIF-1α in the inner zones of the stem-like cells cultured under normoxia suggestive of a hypoxic state in the innermost parts of the oncospheres (Figure 3A). All our stem-like cells expressed HIF-2α under both microenvironments, but the expression was upregulated under hypoxia (Figure 3D–F). HIF-2α has been reported to be elevated in glioblastoma stem cells irrespective of the microenvironment [40,41], which was also evident from our results.

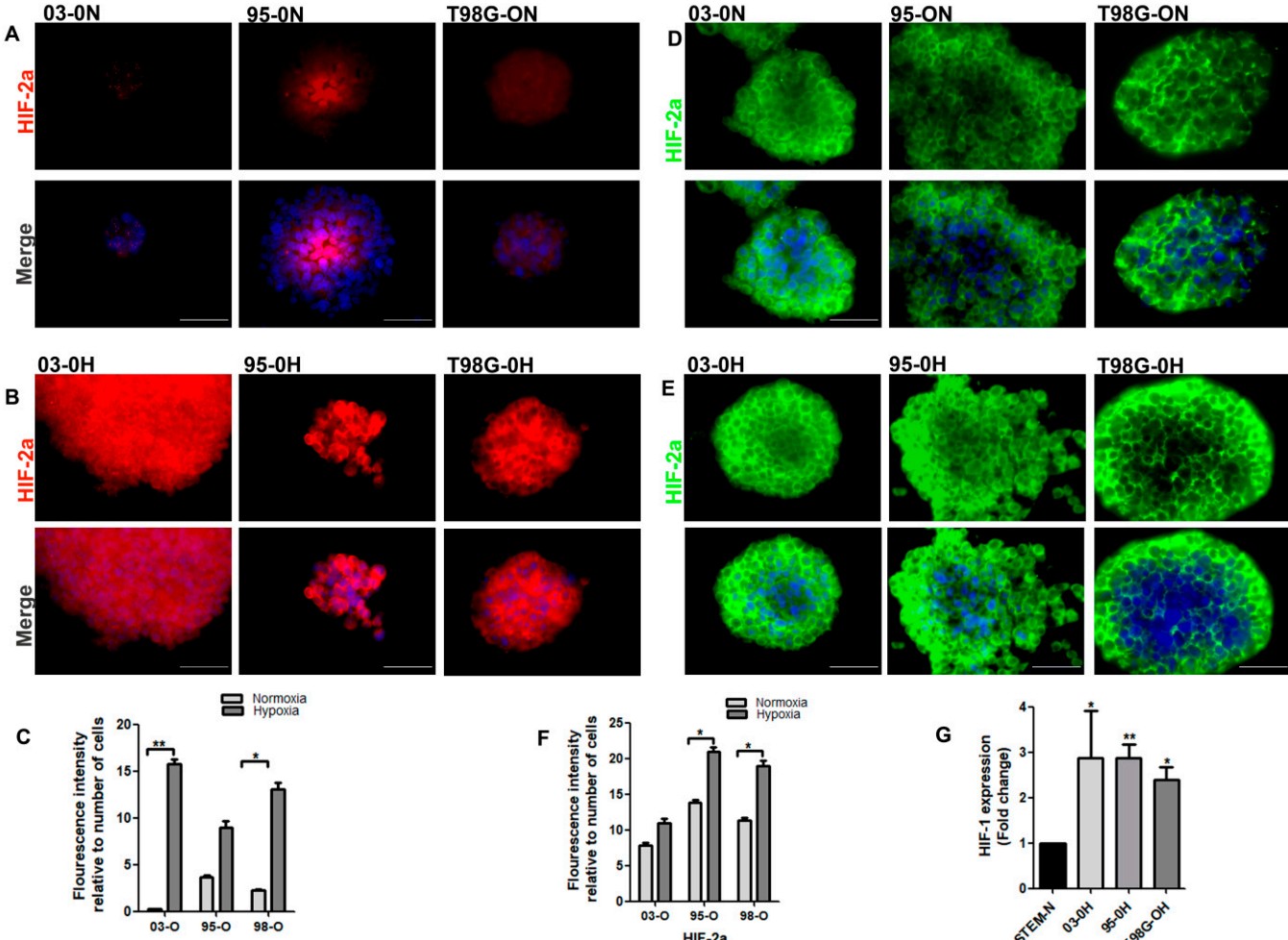

**Figure 3.** Expression of HIF-1α and HIF-2α before and after the hypoxia culture. The stem-like cell lines (03-O, 95-O, and T98G-O) were cultured under a normoxia (N) or a hypoxia (H) microenvironment for 72 h. The cells were evaluated for the expression of HIF-1α after normoxia (**A**) or hypoxia culture (**B**) and for the expression of HIF-2α after normoxia (**D**) or hypoxia culture (**E**) by immunofluorescence. (**C,F**) The quantification graphs of HIF-1/2α. (**G**) Shows the expression of HIF-1α by qRT-PCR where the relative fold change difference was calculated using 2-$^{ΔΔ}$CT method with ACTIN as the normalizer. Each value represents the mean ± SD of three independent experiments where, * $p < 0.05$, ** $p < 0.01$,. Scale bar = 50 μm.

### 3.3.1. Effect of Hypoxia on the Expression of miRNAs in the Stem-Like Cells

Hypoxia has been shown to influence the expression of miRNAs as highlighted in our previous article [15]. Regardless, there are limited studies evaluating the influence of hypoxia using actual stem cells. To address this, we evaluated the expression of selected miRNAs in our stem-like cells (03-0, 95-0, and T98G-0) cultured under a hypoxia microenvironment. When a comparison was made between hypoxia and normoxia microenvironments, we were able to see a downregulation with a fold of about 2.5 to 6 for miR-34-5p (Figure 4A), 2 to 5 for miR-128a-3p (Figure 4B) and 2 to 4 for miR-181a-3p

(Figure 4C). We also saw an upregulation fold difference of about 3 to 4 for miR-221-3p (Figure 4D) and 2 to 4 for miR-17-5p (Figure 4E). Hypoxia microenvironment influenced a further downregulation of miRs-34-5p, 128-3p and, 181 and a further upregulation of the miRs-17-5p and 221-3p.

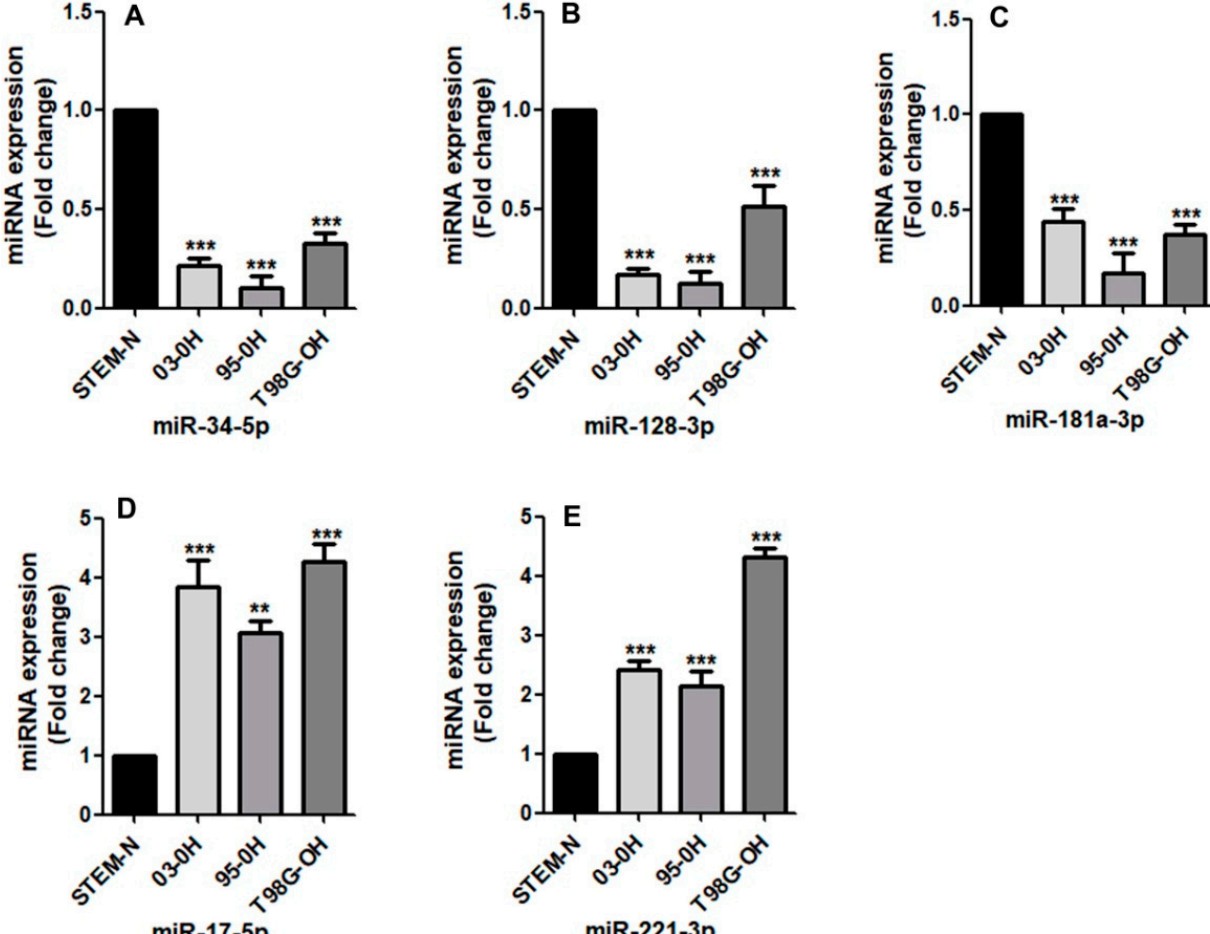

**Figure 4.** Effect of hypoxia on the expression of miRNAs in the stem-like cells. The stem-like cells (03-O, 95-O, and T98G-O) were cultured under normoxia (N) and hypoxia (H) conditions and the expression of miR-34a-5p (**A**), miR-1281a-3p (**B**), miR-181a (**C**), miR-17-5p (**D**) and miR-221-3p (**E**) evaluated by qRT-PCR. The relative fold change difference was calculated using 2-$^{\Delta\Delta}$Cq method, where RNU6 was used as the normalizator. Each value represents the mean $\pm$ SD of three independent runs where, ** $p < 0.01$, *** $p < 0.001$; compared to the stem-like cells under normoxia.

### 3.3.2. Hypoxia Promotes the Stemness Property of the GSCs

The stem-like cells have the ability to express transcription factors including SOX2 and OCT4 [33]. Similarly, the hypoxia microenvironment has been shown to drive the reprogramming of cells towards a stem-cell phenotype and to aid in their maintenance [6,9]. In this manner, we compared the expression levels of SOX2 and OCT4 in our stem-like cells cultured under normoxia and hypoxia microenvironments. There was an upregulated expression of SOX2 (Figure 5A–C) and OCT4 under hypoxia when compared to normoxia (Figure 5D–F). Additionally, the expression of SOX2 and OCT4 was suggestively stronger in the inner zones of the neurospheres cultured under normoxia (Figure 5A,D). The zones were less visible in the stem-like cells cultured under hypoxia where the expression of SOX2 and OCT4 was uniformly distributed in the oncospheres and upregulated. This observation suggests that the hypoxia microenvironment favors the stem cells.

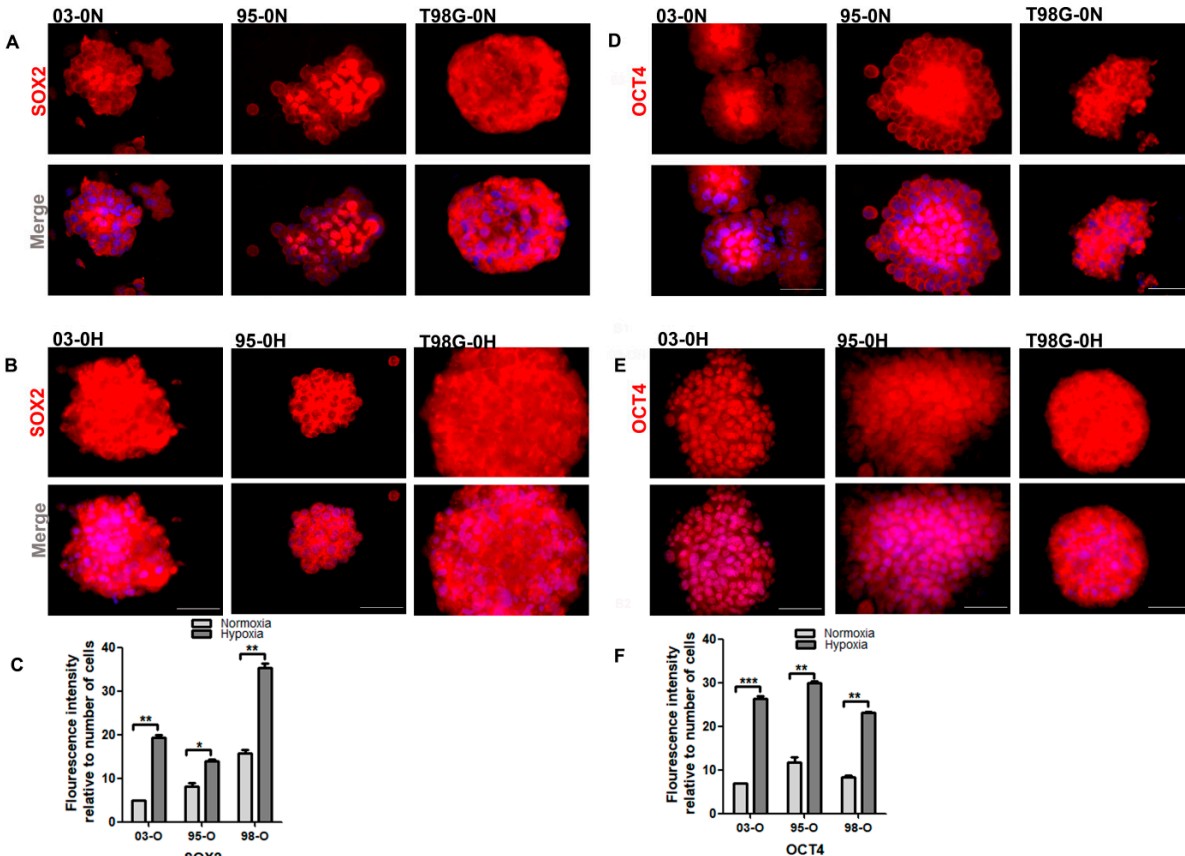

**Figure 5.** Expression of SOX2 and OCT4 under normoxia and hypoxia microenvironments. The stem-like cells (03-O, 95-O, and T98G-O) were cultured under normoxia (N) and in hypoxia (H) for 72 h and stained for SOX2 (**A,B**) and OCT4 (**D,E**) by immunofluorescence. (**C,F**) The quantification graphs of SOX2 and OCT4 respectively. Each value represents the mean ± SD of three independent experiments, where * $p < 0.05$, ** $p < 0.01$, *** $p < 0.001$. Scale bar = 50 μm. The stem-like cells expressed SOX2 and OCT4, but the expression under normoxia was strongest in the inner zones of the neurospheres.

### 3.3.3. Evasion of Apoptosis and Metabolic Reprogramming of the Stem-Like Cells

Resisting cell death and metabolic reprogramming are among the hallmarks of cancer [42,43]. Therefore, we evaluated the effect of hypoxia on the expression of the anti-apoptotic proteins (Bcl-2 and survivin) and GLUT-1 associated with metabolism. Our results showed that our stem-like cells expressed Bcl-2, survivin, and GLUT-1 under a normoxia microenvironment (Figure 6A,C,E), but the expression was upregulated under the hypoxia microenvironment (Figure 6B,D,F,G–I). These results were suggestive of an increased apoptotic escape and altered metabolism in our stem-like cells and these changes were further influenced by the hypoxia microenvironment.

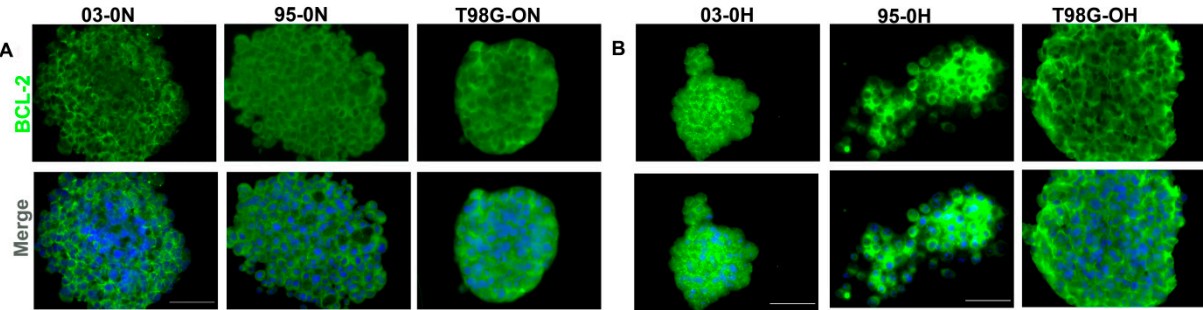

**Figure 6.** *Cont.*

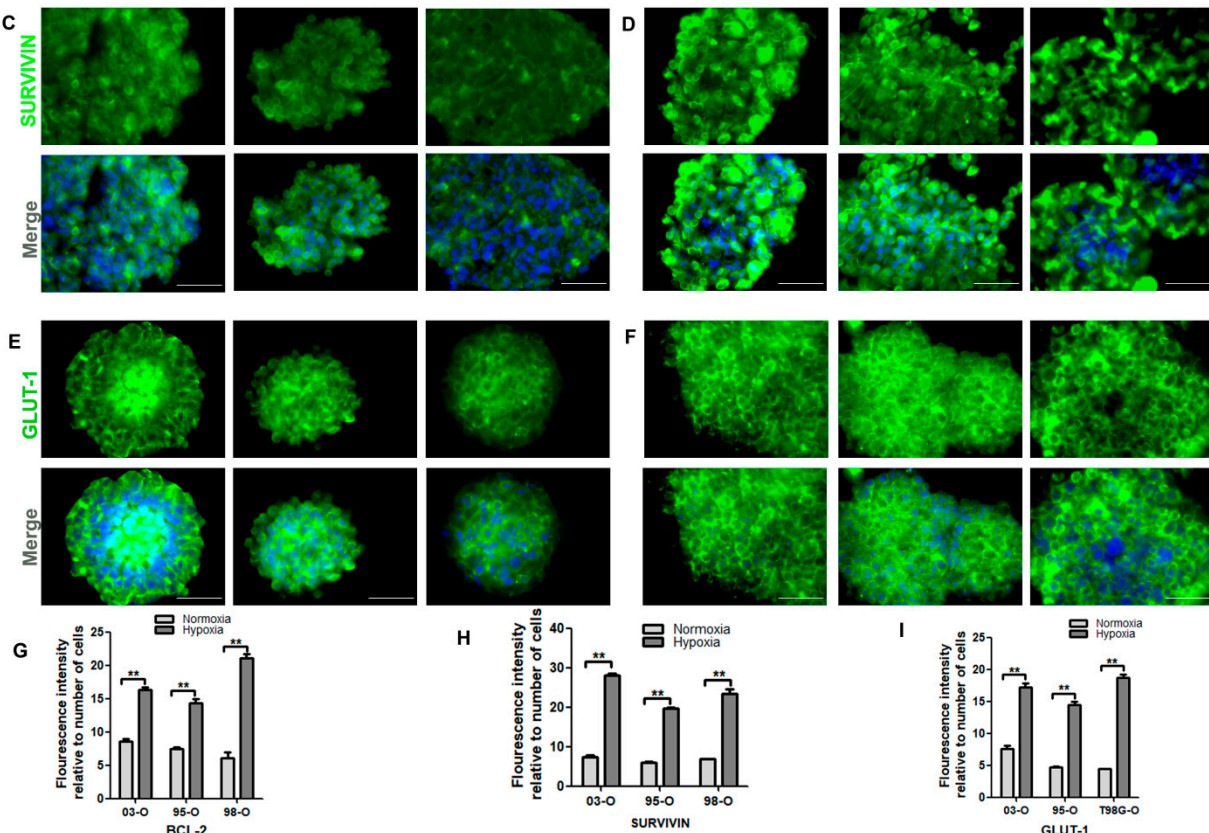

**Figure 6.** Expression of Bcl-2, Survivin and Glut-2 between normoxia and hypoxia microenvironments. The stem-like cells (03-O, 95-O, and T98G-O) were cultured under normoxia (N) and in hypoxia (H) for 72 h and stained for Bcl-2 (**A**,**B**) survivin (**C**,**D**) and Glut-1 (**E**,**F**) by immunofluorescence. (**G**–**I**) The quantification graphs of Bcl-2, survivin and glut-1, respectively. The images were taken with a DMi8 Leica microscope and prepared and quantified using ImageJ. Each value represents the mean $\pm$ SD of three independent experiments, where ** $p < 0.01$. Scale bar = 50 μm.

### 3.4. The 3D Culture Model as a Culture Alternative That Can Mimic the Tumor Microenvironment

The 3D culture system is a closer and more economical culture model that can be used to recapitulate the in vivo tumor microenvironment [25,28]. Therefore, we plated GBM03 and T98G parental cells in a modified 3D culture model made of 24- or 96-well plates coated with agarose and followed their growth for up to 12 days. We compared the spheroids' growth between the conventional culture medium (DMEM with 10% FBS) and the stem cell medium (NS34). Our results showed that both the GBM03 and the T98G cell lines were able to grow as spheroids as early as the fourth day and the spheroids were fully formed by the twelfth day (Figure 7). The choice of the culture medium did not induce visible morphological changes. For the spheroids cultured in 96-well plates, the spheroids cultured using the NS34 medium were suggestively larger in diameter when compared to those grown using DMEM/FBS. However, the spheroids in DMEM/FBS had rounded edges (Figure 7A,B). For the spheroids cultured in 24-well plates, most of the spheroids cultured in DMEM/FBS medium had a slightly larger diameter than those grown under NS34 medium. Regardless, the spheroids in 24-well plates shared similar sphere morphology (Figure 7C,D). We also evaluated the expression of some selected genes (HIF-2α, Glut-1, surviving, and OCT4) in our spheroids by immunofluorescence. Our spheroids expressed these genes in both the culture medium with no significant differences except for HIF-2α and surviving, which were suggestively upregulated in the spheroids cultured using the NS34 medium for both cell lines. Similarly, our spheroids had an outstanding staining pattern highlighting an intricate cellular matrix connection (Figure S3).

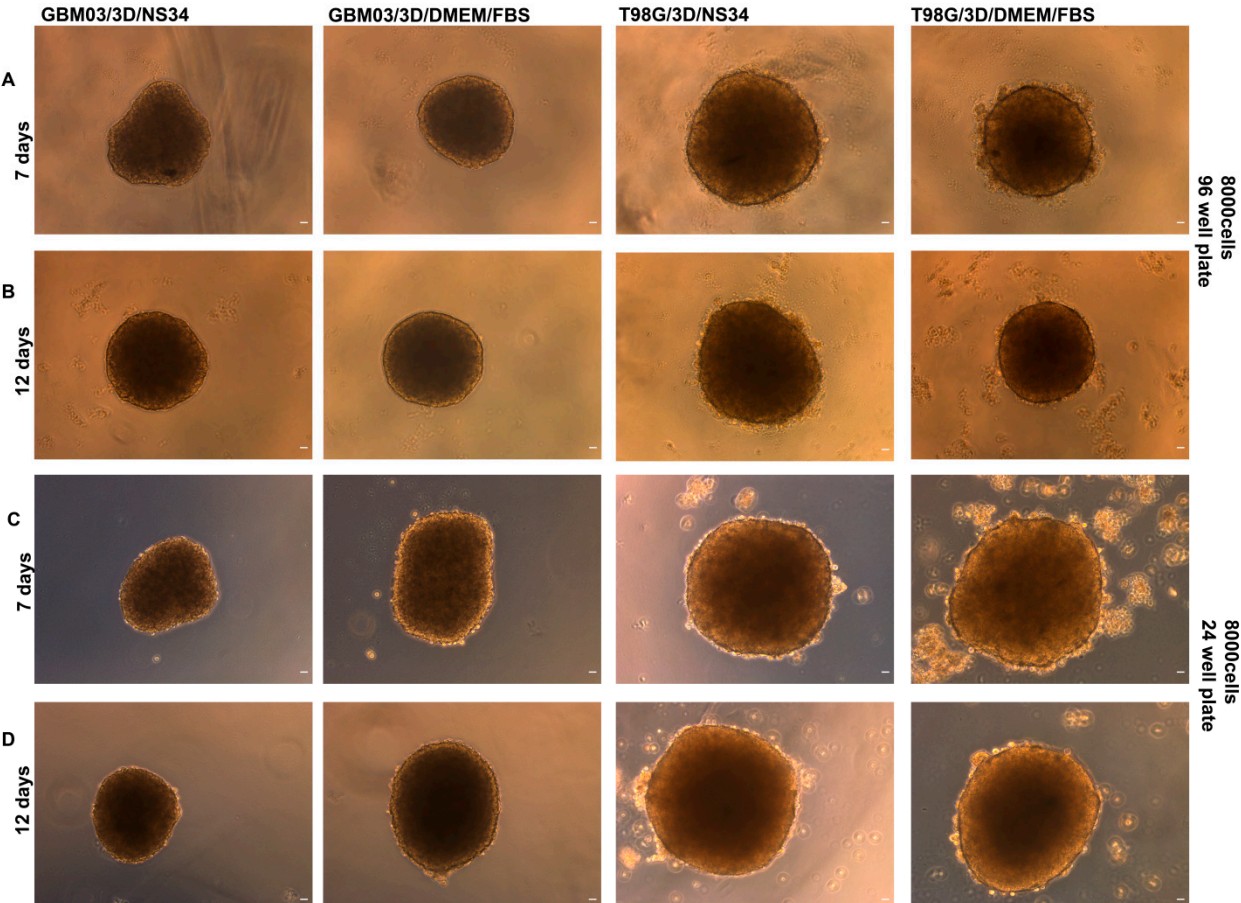

**Figure 7.** Morphological changes in the glioblastoma spheroids. The GBM 03 and T98G parental cells were cultured in 96-well (**A,B**) and 24-well (**C,D**) plates in either NS34 or DMEM/FBS in a 3D modified culture model for a period of 12 days. The photos were taken using a DMi8 Leica microscope and edited using image J. Scale bar = 50 μm. Each experiment was done at least 3 times.

## 4. Discussion

The interest in the role of CSCs in GBM persists due to their involvement in drug resistance and relapse [12]. GSCs are a heterogeneous population present in both the intratumoral perivascular and the necrotic/hypoxic niches [6]. These niches help to maintain the stemness property, regulate proliferation and self-renewal, and to protect the GSCs from environmental insults [10,44]. An improved understanding of the molecular regulators including miRNAs and microenvironmental roles in tumor stem cells is crucial for designing new treatment strategies.

Our study found miRNAs 34-5p, 128-3p, 181a-3p to be downregulated, while miRNAs 221-3p and 17-5p were upregulated when a comparison was made between the stem-like and their parental cells. This expression pattern was similar to the pattern we observed when comparing the parental GBMs and the non-GBM cell line in our previous study [45]. In support of this observation, other sources have reported that miRNA profiles in stem cells are similar to those observed in cancer cells [46]. However, there are limited studies showing miRNA expression using glioblastoma stem cells to make an accurate comparison.

There have been many studies reporting the effect of hypoxia on glioblastoma cell lines as compared to using actual cancer stem cells [47]. In this regard, we investigated the interaction between GSCs and observed that when the stem-like cells were grown under hypoxic conditions, they were larger in diameter than their normoxic counterparts. Hypoxia has already been reported to enhance neurosphere formation and cell proliferation in a study that treated T4121 glioma stem and non-stem cells in hypoxia for several days while noting their distinct growth advantages. The study showed that cells grown in hypoxia

exhibited increased growth compared to the cells grown in normoxia [48]. We were able to observe from our study that some neurospheres under the normoxia microenvironment had zones via immunocytochemistry results. This observation suggested that the inner zones of some spheres seemed to have higher expression of the markers under study, suggesting a hypoxic situation in the middle of these spheres. In line with this, it has been reported that the presence of low oxygen tensions in stem cell niches offers a selective advantage that is well suited to their particular biological roles [49].

Since most cellular adaptations under hypoxia are commonly regulated by HIFs, we observed an elevated expression of HIF-2$\alpha$ in our stem-like cells grown under normoxia and hypoxia, while HIF-1$\alpha$ was significantly expressed under hypoxia. In line with this, it has been reported that HIF-2$\alpha$ is preferentially expressed by CD133+ putative GSCs under both hypoxic and normoxic conditions, whereas HIF-1$\alpha$ is induced under hypoxia not only in CD133+ but also in CD133$^-$ glioma cells [40]. This evidence supports the idea that HIFs play a role in cellular adaptations of the stem-like cells under hypoxia.

When we compared the expression of miRNAs in our stem-like cells cultured under hypoxia to those cultured under normoxia, we were able to see a downregulation with a fold of about 2.5 to 6 for miR-34-5p, 2 to 5 for miR-128a-3p, and 2 to 4 for miR-181a-3p. We also saw an upregulation fold difference of about 3 to 4 for miR-221-3p and 2 to 4 for miR-17-5p. There are few pieces of evidence elucidating the underlying effect of hypoxia on miRNAs using actual GSCs as most evidence is based on glioblastoma cells. The miR-17-92 cluster has been reported to be reduced in hypoxia-treated cells. The downregulation of miR-17-92 by hypoxia resulted in the stabilization of HIF1, because this HRM is able to repress the expression of HIF1 through direct targeting [50]. The miR-128 can modulate angiogenesis, functioning through suppression of *P70S6K1*, a kinase upstream of HIF-1a and VEGF. Induced expression of miR-128 is able to attenuate these effects, while forced expression of *P70S6K1* can partly rescue the inhibitory function of miR-128 on cancer growth [51]. Elsewhere, a study that performed miRNA profiling in GBM spheroid cultures grown in either 2% or 21% oxygen found miR-210 to be significantly upregulated in hypoxia in patient-derived spheroids [52]. These pieces of evidence show that miRNAs may represent a therapeutic target, although it is not clear from the results whether this miRNA may be related to specific cancer stem cell functions.

Besides HIFs, the expression of SOX2, OCT4, GLUT-1, BCL-2, and survivin were also found to upregulated in the stem-like cells under normoxia and even further upon hypoxia induction. This observation could be attributable to the transcription factor HIF, which has been reported to regulate the expression of these genes [44,53,54]. Restricted oxygen conditions have also been reported to expand the fraction of cells positive for a cancer stem cell marker or the side population in established cancer cell lines [55,56]. Notably, Oct4 and Sox2 are already confirmed factors that contribute to the survival and self-renewal of brain tumor stem cells [57,58]. Cancer stem cells have also been shown to regulate Glut1, a transcriptional target of HIF, under hypoxia to a greater degree than non-stem cells [40]. Elsewhere, a study found Bcl-2 mRNA and its protein to be highly expressed in brain glioma stem cells when compared to their corresponding primary glioma cells, and that as an anti-apoptotic gene, Bcl-2 assigns immortality characteristics to cells [59,60]. Upregulation of survivin has also been observed in well-characterized patient-derived GSC lines, and has been shown to contribute to therapy resistance in GSCs [61]. Our data showed the stem-like state and the hypoxia microenvironments to be influencing the expression of the selected genes.

The 3D culture system can provide a simple realistic model for culturing spheres that recapitulate the microenvironmental aspects creating the cell niches that can be used to study cellular interactions in depth. When we cultured our GBM03 and T98G cell lines under our modified 3D culture system, all our cell lines were able to form spheroids by the twelfth day. The spheroid morphology was not so different between the cells cultured in NS34 and those cultured in DMEM/F12 with serum. The spheroids grown in both culture mediums expressed HIF-2$\alpha$, Glut1, survivin, and OCT4 by immunocytochemistry, except

for HIF-2$\alpha$ and survivin, which were suggestively upregulated in the spheroids cultured in NS34 media. Similarly, these spheroids had outstanding staining patterns highlighting an intricate cellular matrix connection. A study that compared the expression of OCT4 between 2D, neurosphere, and 3D models found OCT4 to be upregulated in the 3D assays compared to the other two models [25]. A study using MCF-7 found the cells to exhibit breast cancer stem-like properties, as shown by their high expression of SOX2 and OCT4 when cultured as spheroid cells as compared to monolayer culture [62]. Subject to further confirmation, this evidence suggests that a 3D culture model could potentially be used as an cheaper alternative culture model. This is because the culture of neurospheres is very expensive and time-consuming, compared to spheroid culture. Neurospheres take about four weeks to be fully formed, while the spheroid culture takes less than twelve days. Both culture models result in cell aggregation that is able to capture the hypoxia microenvironment possibly in the inner zones, but these assumptions need to be investigated further.

## 5. Conclusions

Our results suggest that the hypoxia microenvironment played a role in supporting the growth of our stem-like cells. This is shown by the increased neurosphere size and the upregulation of HIF-1/2, SOX2, OCT4, VEGF, GLUT-1, BCL2, and survivin under hypoxia. These genes are involved in the regulation of stemness, metabolism, angiogenesis, and anti-apoptotic properties, all of which are important cancer hallmarks. Our results also showed a differential expression of miRNAs between the stem-like cells and the GBMs, and that the hypoxia microenvironment influenced further dysregulation of the same miRNAs. The miRNAs may be promising therapeutic targets against the GSCs, which are a limiting factor to finding an effective cure for glioblastoma. However, more research is needed to conclusively demonstrate the effect of the hypoxia microenvironment on the GSCs. Our cells were also able to grow as spheroids, and maybe the use of a 3D culture model can serve as a cheaper alternative for capturing the in vivo tumor microenvironment. In summary, we support that the hypoxia microenvironment (1% oxygen) plays a role in the stem-like state, and that it is important to consider when doing experiments, as this microenvironment influences the growth of oncospheres, the expression of miRNAs, and the expression of the tumor-associated genes. Additionally, targeting the hypoxic niche may prove an effective anti-cancer treatment.

**Supplementary Materials:** The following supporting information can be downloaded at: https://www.mdpi.com/article/10.3390/onco2020008/s1, Figure S1: Conversion of GBM cells into stem-like cells; Figure S2: Morphological changes of stem-like cells under normoxia and hypoxia microenvironments; Figure S3: Expression of selected genes by the glioblastoma spheroids.

**Author Contributions:** The article was conceptualized, written, edited, and critically evaluated by each of the authors. The final preparation was done by L.W.M. The work was supervised by V.M.-N. while the revision of the manuscript was done by T.C.L.S.S. Funding acquisition was done by V.M.-N. All authors have read and agreed to the published version of the manuscript.

**Funding:** This study was supported by the Brazilian agencies Conselho Nacional de Desenvolvimento Científico e Tecnológico (CNPq), Coordenação de Aperfeiçoamento de Pessoal de Nível Superior (CAPES), Fundação de Amparo à Pesquisa do Rio de Janeiro (FAPERJ) and Ary Frauzino Foundation for Cancer Research for their support.

**Institutional Review Board Statement:** This study was approved by the Ethics Committee at the Center for Health Sciences at the Federal University of Rio de Janeiro (Protocol No. DAHEICB 015) and by the Brazilian Ministry of Health Ethics Committee (CONEP Protocol No. 2340).

**Informed Consent Statement:** Not applicable.

**Data Availability Statement:** Most data generated from the study was included in the manuscript. However, if more is needed, it will be provided upon reasonable request.

**Acknowledgments:** The author is a mentee of the African Academy of Sciences (AAS) and we are thankful for their endless support and for their continued mentorship. This opportunity was because of the collaboration that exist between AAS and the Brazilian Academy of Sciences and for that we are grateful. The authors would also like to thank the members of the Laboratório de Biomedicina do Cérebro (LBMC) and the Associação Mahatma Gandhi for their support.

**Conflicts of Interest:** The authors declare no conflict of interest.

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
