# Peer review of "Evaluation of miRNA Expression in Glioblastoma Stem-Like Cells: A Comparison between Normoxia and Hypoxia Microenvironment"

_onco, doi:10.3390/onco2020008_

Round 1

Reviewer 1 Report

General

The manuscript focuses on the molecular characteristics of glioblastoma stem cells (GBM) that may influence the aggressiveness, progression, and response to therapy of this malignancy. The authors studied several key molecules (miRNAs) whose expression pattern is characteristic for GBM and other malignancies associated stem cells. Published studies of GBM stem cells in culture are mainly limited to the research of miRNA signatures of cells grown under normal oxygenation.

However, hypoxic conditions better reproduce the actual development and establishment of glioblastoma stem cells at the organism level. Therefore, it is valuable for the field of glioblastoma research to develop a primary cell culture protocol for the establishment and further monitoring of glioblastoma stem cells under hypoxic microenvironment conditions.

The authors of the manuscript reported a variant of such a protocol and further investigated important characteristics of the cell colonies that developed under this protocol. They demonstrated characteristic changes in the expression levels of several miRNAs to distinguish between three cell types 1) parental GBM cells and GBM stem cells derived from parental cells and later grown 2) under hypoxia or 3) under normal oxygen supply. The results may help to establish a link between the miRNA expression signature detected in cells and the survival of GBM patients and/or their response to therapy. It goes without saying that establishing such a link is very important in cancer research.

While the aim of the study is significant for the field of cancer research, the presentation of the results is a weakness of the study, where there is huge potential for improvement.

Graphical presentation

Image labels are very small and often presented in low contrast. Not all letter labels are explained in the figure legends. In some cases, they are mentioned and explained in the main text, but no explanation is given in the legends. In my opinion, manuscript creators should strive for a presentation of the results, where the reader can look at the figures of the article with legends and understand the main ideas of the research before reading the main text. Moreover, in today's world, many readers begin acquainting themselves with a published paper by studying the research article on their mobile phones. It is therefore very important to have large, contrasting labels on the figures. The recently published paper can serve as an example of optimal presentation of figure labels and legends https://www.nature.com/articles/s41586-022-04702-4_reference.pdf

Tables

The authors of the manuscript do not take advantage of summarizing their results in tables. Therefore, some of their presentations of textual results are overloaded with details that are difficult for the reader to comprehend and analyze. For example, it would be interesting to see a table combining the results from Figure 2 and 4.

Text

The authors of the manuscript do not follow the golden rule of writing, according to which one paragraph should communicate one thought. Therefore, the text of their paper has very long paragraphs that are difficult for the reader to absorb and internally structure.

English is also in need of considerable improvement. Modern online translation tools allow researchers to significantly improve written scientific texts without much effort

https://www.deepl.com/translator  . I'm sure that after translating from English to Portuguese and back again, the clarity of the manuscript could be greatly improved. Otherwise, in its present form, the manuscript is very difficult to understand and requires a great deal of effort and time.

I also suggest that authors reexamine the text of the introduction and discussion sections. Sometimes a detailed description of the facts already established in published studies is more suited to an Introduction and requires less detail in the Discussion section, in which the logical link between what is known and what is found in the current study must be made more clearly. The connection appears for the reader very vague with too much detail. It may also be possible to present already published data in the form of a summary table.

I am sure that the logical structure of the Results, Discussion and most important Conclusion section could be further improved. The authors should study the text looking for phrases and thoughts that are poorly logically connected to each other and strengthen this connection. Further multiple comments, suggestions and corrections can be found in the attached file.

Author Response

 Authors’ Response to Reviewers’ Comments:

Journal: Onco

Manuscript ID: 1694423

Manuscript Title: Evaluation of miRNA expression in glioblastoma stem-like cells: A comparison between normoxia and hypoxia microenvironment

Dear Editor:

We thank the Editorial committee and the reviewers for reviewing our manuscript and for the suggestions. We have revised the manuscript and incorporated all the suggestions. The information about the origin of the cell lines has been added. The introduction, methods, results and discussion sections have been modified according to the suggestions provided. The institutional emails are; LWM [email protected], WM [email protected], JMCA [email protected]., and VMN [email protected]. We hope that this new version of the manuscript is fit for approval and publication at Onco. Kindly find below the point-by-point response to the comments and queries. Please, you find attached here my resumed CV.

Sincerely,

Point by Point Response 

Reviewer 1

Overall Comment:

“The manuscript focuses on the molecular characteristics of glioblastoma stem cells (GBM) that may influence the aggressiveness, progression, and response to therapy of this malignancy. The authors studied several key molecules (miRNAs) whose expression pattern is characteristic for GBM and other malignancies associated stem cells. Published studies of GBM stem cells in culture are mainly limited to the research of miRNA signatures of cells grown under normal oxygenation.

However, hypoxic conditions better reproduce the actual development and establishment of glioblastoma stem cells at the organism level. Therefore, it is valuable for the field of glioblastoma research to develop a primary cell culture protocol for the establishment and further monitoring of glioblastoma stem cells under hypoxic microenvironment conditions.

The authors of the manuscript reported a variant of such a protocol and further investigated important characteristics of the cell colonies that developed under this protocol. They demonstrated characteristic changes in the expression levels of several miRNAs to distinguish between three cell types 1) parental GBM cells and GBM stem cells derived from parental cells and later grown 2) under hypoxia or 3) under normal oxygen supply. The results may help to establish a link between the miRNA expression signature detected in cells and the survival of GBM patients and/or their response to therapy. It goes without saying that establishing such a link is very important in cancer research.

While the aim of the study is significant for the field of cancer research, the presentation of the results is a weakness of the study, where there is huge potential for improvement.” 

Response:

We thank the reviewer for reading our manuscript and for the positive feedback and for the suggestions. We have modified the manuscript and improved on the presentation of the results. Kindly find the point by point response to the comments.

Comments:

  1. Graphical presentation

Image labels are very small and often presented in low contrast. Not all letter labels are explained in the figure legends. In some cases, they are mentioned and explained in the main text, but no explanation is given in the legends. In my opinion, manuscript creators should strive for a presentation of the results, where the reader can look at the figures of the article with legends and understand the main ideas of the research before reading the main text. Moreover, in today's world, many readers begin acquainting themselves with a published paper by studying the research article on their mobile phones. It is therefore very important to have large, contrasting labels on the figures. The recently published paper can serve as an example of optimal presentation of figure labels and legends https://www.nature.com/articles/s41586-022-04702-4_reference.pdf

Response:

We thank the reviewer for the suggestion. We have modified the image labels and improved their contrasts. We have included the explanation of the labels in the figure legends. We are thankful to the reviewer for the observation and for providing us with a template article to serve as an example to follow.

  1. Tables

“The authors of the manuscript do not take advantage of summarizing their results in tables. Therefore, some of their presentations of textual results are overloaded with details that are difficult for the reader to comprehend and analyze. For example, it would be interesting to see a table combining the results from Figure 2 and 4.

Response: We thank the reviewer for the comment. We have improved on the presentation of the results in figure 2 and 4 in a way that is easy to comprehend.

  1. Text

“The authors of the manuscript do not follow the golden rule of writing, according to which one paragraph should communicate one thought. Therefore, the text of their paper has very long paragraphs that are difficult for the reader to absorb and internally structure.”

Response: We thank the reviewer for the observation. We have modified the introduction, methods, results and discussion sections of the new manuscript taking into consideration all the suggestions provided. We are thankful for the suggestions.

“English is also in need of considerable improvement. Modern online translation tools allow researchers to significantly improve written scientific texts without much effort

https://www.deepl.com/translator  . I'm sure that after translating from English to Portuguese and back again, the clarity of the manuscript could be greatly improved. Otherwise, in its present form, the manuscript is very difficult to understand and requires a great deal of effort and time.”

Response: We thank the reviewer for the suggestion. We have edited the manuscript and improved on the grammar to make the new manuscript readable and easy to understand. We are grateful for the observation.

“I also suggest that authors reexamine the text of the introduction and discussion sections. Sometimes a detailed description of the facts already established in published studies is more suited to an Introduction and requires less detail in the Discussion section, in which the logical link between what is known and what is found in the current study must be made more clearly. The connection appears for the reader very vague with too much detail. It may also be possible to present already published data in the form of a summary table.”

Response: We thank the reviewer for the suggestion. We have re-examined the introduction and discussion sections in the new manuscript. We have moved part of the write-up from the discussion section to the introduction as suggested. We are grateful for the suggestion.

“I am sure that the logical structure of the Results, Discussion and most important Conclusion section could be further improved. The authors should study the text looking for phrases and thoughts that are poorly logically connected to each other and strengthen this connection. Further multiple comments, suggestions and corrections can be found in the attached file.”

Response: We thank the reviewer for the observation. We have improved on the logical structure of the results, discussion and conclusion sections of the new manuscript. We have modified the connecting statements or phrases to clarify their intended meaning. We have also modified the new manuscript to include all the suggestions and corrections suggested in the attached file. We are grateful to the reviewer for highlighting the suggestions in the attached file.

Reviewer 2 Report

This research article reports on an investigation of miRNA expression in glioblastoma stem-like cells (GSCs), which is interesting and covers a very relevant topic, especially since it compares normoxic and hypoxic micro-environments. Based on its topic and contents, this investigation warrants publication, but its presentation, the clarity of the materials & methods and other issues in the text currently preclude an efficient communication of its results.  

The major point for improvement is the quality of the images. Confocal imaging is really required to obtain clear images of 3D GSC spheorids. At present, all spheroid images are blurry. This is not acceptable for publication and also prevent this reviewer’s critical appraisal of the images and whether or not the conclusions of this study are supported by its results. Please improve the quality of the images by using confocal imaging.

Other points that require improvement:

  • Introduction:
    • The references are not at all up-to-date (3 references are from 2019, all other references are older). Please add recent publications in the manuscript regarding GSCs, GSC niches and 3D GSC spheroid cultures. I will add some suggestions below.
    • Explain in the introduction why GSCs are resistant to therapy (lines 38-39): this has recently been reviewed in doi: 10.1097/WCO.0000000000000994, doi: 10.3390/ijms22083863 and doi: 10.1186/s13287-021-02231-x
    • In the introduction, you mention that GSCs reside in hypoxic niches (lines 39-40). Please include references that have shown GSCs in hypoxic niches in glioblastoma. Some primary data that supports this finding was published in doi: 10.1369/0022155415581689 and doi: 10.1369/0022155417749174.
  • Materials and methods:
    • How were differentiated cells separated from undifferentiated cells (line 90)?
    • Which transcription factors were evaluated after 15 days in culture medium (lines 90, 91)?
    • Section 2.2: which cell lines were used for the 3D cultures?
    • Line 99: Why was FBS added to the medium for 3D GSC cultures? Serum is known to induce cell differentiation.
    • Section 2.3: Why were 96-wells included without any cells (line 106)?
    • What kind of information was collected and how was this performed (line 110,111)?
    • Section 2.4: How did the spheroids adhere to the coverslip?? How were the cells permeabilized? What reagent was used to stain for the cellular nuclei?

Confocal imaging is required to obtain clear images of 3D structures, such as GSC spheroids.

  • Discussion:
    • Hypoxic conditions versus normoxic conditions were compared to study GSCs. How about co-culturing multiple cell types that are part of the GSC microenvironment? Has this been considered? If yes, what kind of studies are suitable to GSCs in more depth?
    • Can the miRNA profile be used to improve anti-glioblastoma treatment?

Author Response

Authors’ Response to Reviewers’ Comments:

Journal: Onco

Manuscript ID: 1694423

Manuscript Title: Evaluation of miRNA expression in glioblastoma stem-like cells: A comparison between normoxia and hypoxia microenvironment

Dear Editor:

We thank the Editorial committee and the reviewers for reviewing our manuscript and for the suggestions. We have revised the manuscript and incorporated all the suggestions. The information about the origin of the cell lines has been added. The introduction, methods, results and discussion sections have been modified according to the suggestions provided. The institutional emails are; LWM [email protected], WM [email protected], JMCA [email protected]., and VMN [email protected]. We hope that this new version of the manuscript is fit for approval and publication at Onco. Kindly find below the point-by-point response to the comments and queries. Please, you find attached here my resumed CV.

Sincerely,Vivaldo Moura Neto

Reviewer 2:

Overall Comment:

“This research article reports on an investigation of miRNA expression in glioblastoma stem-like cells (GSCs), which is interesting and covers a very relevant topic, especially since it compares normoxic and hypoxic micro-environments Based on its topic and contents, this investigation warrants publication, but its presentation, the clarity of the materials & methods and other issues in the text currently preclude an efficient communication of its results.”

Response: We thank the reviewer for revising our manuscript and for the positive feedback and the comments. We have improved on the presentation of the manuscript especially the material and methods sections as suggested. Kindly find the point by point response to the comments below.

The major point for improvement is the quality of the images. Confocal imaging is really required to obtain clear images of 3D GSC spheorids. At present, all spheroid images are blurry. This is not acceptable for publication and also prevent this reviewer’s critical appraisal of the images and whether or not the conclusions of this study are supported by its results. Please improve the quality of the images by using confocal imaging.

Response: We are thankful to the reviewer for the suggestion. We have improved on the images by presenting their original TIFF formats We have also removed the image labels that were crowding most of the images. The authors are not in a position to use confocal imaging option because the study came to an end. In our humble justification, we used the oil immersion lens of the DMI8 leica microscope that has a super resolution to read the images. However, the 3D morphology of the clls made it difficult for the oncospheres and spheroids to incorporate the triton and the markers efficiently especially if the spheres were large.

Other points that require improvement:

  1. Introduction:

The references are not at all up-to-date (3 references are from 2019, all other references are older). Please add recent publications in the manuscript regarding GSCs, GSC niches and 3D GSC spheroid cultures. I will add some suggestions below.

Explain in the introduction why GSCs are resistant to therapy (lines 38-39): this has recently been reviewed in doi: 10.1097/WCO.0000000000000994, doi: 10.3390/ijms22083863 and doi: 10.1186/s13287-021-02231-x

In the introduction, you mention that GSCs reside in hypoxic niches (lines 39-40). Please include references that have shown GSCs in hypoxic niches in glioblastoma. Some primary data that supports this finding was published in doi: 10.1369/0022155415581689 and doi: 10.1369/0022155417749174.

Response: We thank the reviewer for the suggestions. We have updated the references to include the suggestions provided above and we are grateful to the reviewer for providing us with the references.

  1. Materials and methods:

How were differentiated cells separated from undifferentiated cells (line 90)?

Response: We thank the reviewer for the question. The differentiated cell lines remained adhered to the culture plate surface while the undifferentiated (stem-like cells) grew into oncospheres and detached from the surface of the culture flask. When sufficient spheres had formed, they were aspirated out of the original culture plates and transferred in new culture plates where they continued to grow as oncospheres. We have modified the manuscript to include this explanation.

Which transcription factors were evaluated after 15 days in culture medium (lines 90, 91)?

Response: We thank the reviewer for the question. We evaluated for the expression of the transcription factors SOX2 and OCT4 associated with stemness. We have modified the manuscript to include the factors.

Section 2.2: which cell lines were used for the 3D cultures?

Response: We thank the reviewer for the question. We used GBM03 and T98G for the 3D cultures. We have modified the manuscript to include this information.

Line 99: Why was FBS added to the medium for 3D GSC cultures? Serum is known to induce cell differentiation.

Response: We are thankful for the question. We used DMEM/F12 with FBS or NS34 without FBS for the 3D cultures. We wanted to compare the GBM spheroids grown with normal media as used in the protocol described by Tang and Chen, 2016 with the spheroids grown with the NS34 stem-cell medium. Because of the agarose, the cells do not adhere and instead grow as spheroids even in the presence of FBS.   

Section 2.3: Why were 96-wells included without any cells (line 106)?

Response: We thank the reviewer for the question. We have modified the clonogenicity assay section to clarify the protocol. We are thankful for the observation.

What kind of information was collected and how was this performed (line 110,111)?

Response: We thank the reviewer for the question. The information collected was the number of spheres per well during the four weeks. An average of the number of spheres was calculated weekly and recorded. The average obtained was used to draw the graph presented in figure 1. We have modified the clonogenicity assay section of the new manuscript to clarify this information.

Section 2.4: How did the spheroids adhere to the coverslip?? How were the cells permeabilized? What reagent was used to stain for the cellular nuclei?

Response: We thank the reviewer for the question. The coverslips were treated with polylysine or polyornithine that helped the oncospheres and spheroids to adhere to the coverslip. The cells were permeabilized with 0.1-0.5% triton depending on the antibody and Dapi was used to stain the nucleus.

Confocal imaging is required to obtain clear images of 3D structures, such as GSC spheroids.

Response: We are thankful for the comment. We have improved the quality and presentation of the images obtained by DMI8 leica microscope. The authors agree with the reviewer’s suggestion but are not in a position to use confocal imaging at this point of the study. In our humble justification, the 3D morphology of the spheroids and neurospheres interfered with proper uptake of the triton and the markers reducing the clarity in a few of them. However, we have modified the images to improve on their presentation.

  1. Discussion:

Hypoxic conditions versus normoxic conditions were compared to study GSCs. How about co-culturing multiple cell types that are part of the GSC microenvironment? Has this been considered? If yes, what kind of studies are suitable to GSCs in more depth?

Response: We thank the reviewer for the comment. For the purpose of this study, we were only able to culture GSCs from GBM03, 95 and T98G glioblastoma cell lines. 

Can the miRNA profile be used to improve anti-glioblastoma treatment?

Response: We thank the reviewer for the question. miRNAs can be used as therapeutic targets for the treatment of glioblastoma and some of them like miR-34 are already in clinical trials.
